# The Roles of Peptide Hormones and Their Receptors during Plant Root Development

**DOI:** 10.3390/genes12010022

**Published:** 2020-12-25

**Authors:** Yu-Chun Hsiao, Masashi Yamada

**Affiliations:** 1Agricultural Biotechnology Research Center, Academia Sinica, Taipei 115, Taiwan; courage241@gate.sinica.edu.tw; 2Biotechnology Center in Southern Taiwan, Academia Sinica, Tainan 711, Taiwan

**Keywords:** root meristem, CLE peptide, RGF1 peptide, PSK peptide, RALF peptide, ROS, LRR-RLK

## Abstract

Peptide hormones play pivotal roles in many physiological processes through coordinating developmental and environmental cues among different cells. Peptide hormones are recognized by their receptors that convey signals to downstream targets and interact with multiple pathways to fine-tune plant growth. Extensive research has illustrated the mechanisms of peptides in shoots but functional studies of peptides in roots are scarce. Reactive oxygen species (ROS) are known to be involved in stress-related events. However, recent studies have shown that they are also associated with many processes that regulate plant development. Here, we focus on recent advances in understanding the relationships between peptide hormones and their receptors during root growth including outlines of how ROS are integrated with these networks.

## 1. Introduction

Small peptides are important signaling molecules that coordinate comprehensive intercellular communication in numerous aspects of developmental processes in plants. The 18-amino acid polypeptide SYSTEMIN was the first functional peptide ligand to be isolated from tomato leaves in 1991 and is involved in the wounding response [1]. From that time on, various peptide ligands have been identified in plants that play versatile roles during developmental processes such as CLAVATA3 (CLV3)/EMBRYO SURROUNDING REGION peptide (CLE), which is associated with stem cell maintenance; PHYTOSULFOKINE (PSK) and PLANT PEPTIDE CONTAINING SULFATED TYROSINE (PSY), which are associated with cell proliferation; ROOT MERISTEM GROWTH FACTOR (herein called RGF, also known as CLE-like or GOLVEN peptide [2,3]), which is required for maintenance of the root stem cell niche and transit-amplifying cell proliferation; CASPARIAN STRIP INTEGRITY FACTORS (CIFs), which are required for Casparian formation, and RAPID ALKALINIZATION FACTOR (RALF), which is involved in calcium-mediated signaling and regulation of root growth [4,5,6,7,8,9].

These peptide hormones usually bind to membrane-localized receptors causing conformational changes and activating downstream signaling to modulate a range of biological functions. The major receptors for peptide signals are the LEUCINE-RICH REPEAT RECEPTOR-LIKE KINASEs (LRR-RLKs) of which there are more than 200 members in *Arabidopsis* [10]. For instance, the well-characterized CLV3 peptides and their receptor CLV1 function together to mediate shoot apical meristem (SAM) maintenance [11]. *CLV3* downregulates the expression of the homeodomain transcription factor *WUSCHEL* (*WUS*) to negatively control the stem cell population [12]. Intriguingly, *WUS* positively regulates *CLV3* expression thus forming a feedback loop to determine the stem cell fate [13]. However, relatively fewer peptide-receptor pairs are characterized in the root. As plants are sessile, roots are fundamental organs that anchor plants in the ground but also absorb water and nutrients from the soil for plant survival. Root tissues are produced by the root apical meristem (RAM). The *Arabidopsis* primary root development can be defined into three major developmental zones (Figure 1): (a) the meristematic zone, where cells actively proliferate, (b) the elongation zone where cells stop proliferation and start elongation, and (c) the differentiation zone where cells differentiate terminally and have acquired their destined fates [14]. The spatiotemporal balance between stem cell maintenance, proliferation, and differentiation determines the root growth rate and is not only mediated by phytohormones but also modulated by small peptides and ROS [4,7,15,16,17,18,19].

In the past, ROS are thought to continuously produced as undesirable byproducts of different metabolic pathways that are localized in several cellular compartments such as chloroplasts, mitochondria, and peroxisomes. A common feature of ROS is that they cause oxidative stress thus damaging the cells [20]. However, cell damage is now considered to be the outcome of ROS triggering a programmed cell death pathway instead of result from the ROS toxicity. The levels of ROS are tightly controlled by the balance between production and scavenging that is essential for promoting normal cellular processes [21]. Besides, more and more studies demonstrate that plants also utilize ROS as signaling molecules for regulating development and various physiological responses [22,23]. ROS include a variety of small molecules, for example, consisting of the singlet oxygen that oxidizes lipids, proteins, and DNA; superoxide (O_2_^•–^), which reacts with iron-sulfur proteins. Hydrogen peroxide (H_2_O_2_), which is comparatively stable and can cross plant membranes and is, therefore, considered to be the predominant ROS involved in cell-cell communication [24]. In this review, we focus on understanding the signaling and functions of peptide hormones and their receptors, mostly CLEs, PSKs, PSYs, RGFs, CIFs, and RALFs, during root development, especially in the development of primary roots and lateral roots including the roles of ROS in these physiological processes.

## 2. The Role of CLE Peptides and Their Receptors during Root Development

Currently, 32 *CLE* genes, which encode 27 distinct CLE precursor proteins with a conserved 12–14 amino acid CLE motif, have been identified in the *Arabidopsis* genome [25]. Secreted CLE peptides induce various intracellular signaling through LRR-RLK during plant development. Overexpression and exogenous supplement experiments are often carried out because of the lack of visible phenotypes of the *cle* null and knockdown mutants in the roots [4,26,27]. The CLE peptides negatively regulate meristem development. Treatment with CLEs results in the termination of the root meristem. Root growth assays supplied with one of 22 synthetic CLE peptides define two functional types of CLEs. Those involved in the inhibition of root meristem development are named A-type CLE peptides and include the well-known CLV3 and other 17 CLE peptides, whereas B-type CLE peptides (CLE41-CLE44) do not affect the RAM growth [26]. To date, there is only one example that shows an interaction between a CLE peptide and receptors in root meristem development [28]. The *cle40* mutant forms additional columella stem cell layers and this phenotype can be rescued by CLE40 treatment. Similarly, both the *clv1* and *arabidopsis crinkly 4* (*acr4*) receptor kinase mutants have excess columella stem cell layers but are not sensitive to CLE40. These two receptors form homo- or hetero-dimers in plants. These findings indicate that CLV1 and ACR4 mediate CLE40 signaling for distal root meristem cell regulation [28], (Figure 1e).

Mutants of the putative A-type CLE peptide receptor genes demonstrate phenotypes resistant to A-type CLE peptides. The *suppressor of overexpression of LLP1 2* (*sol2*)/*coryne* (*crn*) mutant has been isolated as a suppressor mutant of a short-root phenotype by overexpression of the *CLE19* gene [29]. Genetic and molecular analyses show that SOL2/CRN and CLV2 work together in the different pathways of CLV1 in the SAM [30,31]. The roots of the *sol2*/*crn* and *clv2* mutants are resistant to various A-type CLE peptide treatments including CLE19 suggesting that the CLV2-CRN/SOL2 may function as a receptor in roots as well [32], (Figure 1e). However, these *cle* mutants and *sol2*/*crn* and *clv2* receptor mutants do not show obvious root meristem phenotypes.

The *RECEPTOR-LIKE PROTEIN KINASE 2* (*RPK2*, also called *TOADSTOOL 2*) gene has been shown to control cell proliferation in the RAM. The *rpk2* mutants are insensitive to CLE17 and CLE19 peptide treatment, indicating RPK2 might be their receptor or in a complex with CLV2-CRN/SOL2 [33]. The *rpk1* mutants showed a partial root growth arrest phenotype, which leads to variable root length that is enhanced in the heterozygous *rpk2* mutant background, suggesting a dose-dependent requirement for both *RPK1* and *RPK2* in the signaling transduction that regulates root growth and meristem patterning [33]. Together, these findings suggest that SOL2, CLV2, RPK1, and RPK2 function as receptors in root meristem development (Figure 1e).

## 3. The Roles of Tyrosine-Sulfated Peptides and Their Receptors in Root Development

While CLE peptides negatively regulate meristem development, tyrosine-sulfated peptides positively control root meristem development. The PSKs, PSYs, and RGFs rely on tyrosine sulfation, a post-translational modification (PTM) catalyzed by the product of a single gene*, TYROSYL-PROTEIN SULFOTRANSFERASE* (*TPST*), for peptide maturation [34], (Figure 1b and Figure 2a). The *tpst-1* mutant shows pleiotropic phenotypes including severely shortened roots caused by the reduced size of the meristematic zone as well as the RAM, indicating the irreplaceable roles of tyrosine sulfation in plant growth and root meristem development [34]. Exogenous addition of either PSY1 or PSK peptides to the *tpst-1* mutant partially rescues the defects, lending the evidence supporting the roles of PSY1 and PSK in cell proliferation in the meristematic zone [7]. Moreover, PSK contributes to the regulation of quiescent center (QC) cell division and the differentiation of distal stem cells [35]. The application of RGF1 peptide restores root meristem size defects in the cell division zone of the *tpst-1* mutant. Furthermore, the *rgf1,2,3* triple mutant exhibits a short-root phenotype that is reminiscent of the *tpst-1* mutant [7]. Additionally, the root growth defects of *tpst-1* could be fully restored in the presence of RGF1, PSK, and PSY1, suggesting that these three peptides are modified by TPST and required for root meristem development [7], (Figure 2a).

In *Arabidopsis*, PSK signals are perceived by two related LRR-RLKs, predominantly through PHYTOSULFOKINE RECEPTOR 1 (PSKR1) to regulate root and hypocotyl elongation, and mildly through PSKR2 to regulate root but not hypocotyl elongation [36]. PSKR1 also forms a signaling module at the plasma membrane together with CYCLIC NUCLEOTIDE GATED CHANNEL 17 (CNGC17), the H^+^-ATPases AHA1 and AHA2, and BRI1-ASSOCIATED RECEPTOR KINASE 1 (BAK1) to mediate cell expansion [37], (Figure 1b). PSY1, on the other hand, is perceived by the PSY1 RECEPTOR (PSY1R), which functions in plant defense responses [9]. PSY1R phosphorylates and activates AHA1 and AHA2 at Thr-881, resulting in extracellular acidification and cell expansion [38].

Eleven *RGF* genes have been identified in the *Arabidopsis* genome [2,3,7]. *RGF1* is specifically expressed in the QC cells and the columella stem cells in the RAM [7]. The other two homologs *RGF2* and *RGF3*, are mostly expressed in the innermost layer of the columella cells and the QC cells [7]. These peptides function in maintaining the root stem cell niche and regulating meristematic activity [7]. Five receptors of the RGF peptides, which are called RGFRs or RGF1 INSENSITIVEs (RGIs), have been identified by three independent groups using different experimental approaches [39,40,41]. The single mutants of individual RGF receptors showed smaller meristem sizes and fewer meristematic cortex cells compared with the wild-type (WT), and the *rgfr1 rgfr2* and *rgfr3 rgfr4* double mutants displayed more severe defects in the root meristem than each of their single mutants, indicating redundant roles of these receptors in meristem maintenance [40]. The *rgi1,2,3,4,5* quintuple mutant exhibits an extremely short root phenotype, with only a few meristematic cortex cells [39]. The expression levels of *PLETHORA 1* (*PLT1*) and *PLT2* are barely detected in *rgi1,2,3,4,5* quintuple mutants, and ectopic expression of *PLT2* driven by the promoter of *RGI2* fully rescues the meristem size of the *rgi1,2,3,4* quadruple mutant and also largely restores the root meristem defects in the *rig1,2,3,4,5* quintuple mutant [39]. Biochemical analyses demonstrated that RGF1 induced the interaction between RGFR1 or RGFR2 and SOMATIC EMBRYOGENESIS RECEPTOR-LIKE KINASE (SERK1/2) /BAK1 [40], while RGF2-5, 8 only induced the RGFR1 and SERK1 heterodimerization. Consistent with the biochemical results, *serk* and *bak* mutants with different combinations displayed shrinking meristems and shorter roots than WT, suggesting that SERKs are important in RGF1-mediated signaling and act as co-receptors for RGFRs in regulating root meristem development [40].

Recently, two research groups elucidated the downstream signaling cascade of RGF1-RGI1 which regulates root meristem maintenance [42,43]. After RGF1 treatment MITOGEN-ACTIVATED PROTEIN KINASE KINASE 4 (MKK4) and MAP KINASE 3 (MPK3) were identified in the RGI1 complex by mass spectrometry (MS) [42]. Constitutively expressed active *MKK4* or *MKK5* could partially rescue the root defects of the *rgi1,2,3,4,5* quintuple mutant. The root meristem size was significantly reduced in the *mkk4,5* double mutant and *mkk4,5* plants were insensitive to RGF1 treatment [42,43]. In addition, the *rgi1,2,3,4,5 mkk4,5* septuple mutant did not exhibit additive effects in the root meristem phenotype, implying that they are involved in the same pathway [42]. *MPK3* and *MPK6* play redundant roles and function upstream of *PLT1/2* to regulate the mitotic activity in the root meristem [43]. The phosphorylation of MPK3/6 in response to sulfated RGF1 peptide is dependent on RGI and MKK4/5 [42,43]. Genetic and biochemical experiments further revealed that YDA, MKK4/5, and MPK3/6 act as downstream signaling components of RGF1-RGIs to modulate the expression of *PLT1* and *PLT2* for root meristem development [42,43], (Figure 2a). Further study is needed to identify more regulatory components that participate in the RGF1–RGFR signaling pathway that fine-tune the sophisticated root meristem development process.

The reversible ubiquitination-mediated degradation has been shown to be one of the most efficient ways to fine-tune cellular protein abundance. Currently, two closely related ubiquitin-specific proteases (UBPs), UBP12 and UBP13, have been shown to act as positive regulators of root meristem development in *Arabidopsis* [44]. The *ubp12,13* double mutant displayed pleiotropic phenotypes, including reducing the number of meristematic cells in the cortex and growth arrest at an early stage. A series of biochemical analyses showed that UBP13 physically interacts with the cytoplasmic domains of RGFR1 and RGFR2, but not RGFR3. The RGFR1 protein level significantly decreased in the *ubp12,13* plants and the *ubp12,13* double mutant is insensitive to RGF1 treatment, demonstrating that UBP12/13 can counteract RGF1-induced RGFR1 ubiquitination further stabilizing RGFR1, and maintaining root cell sensitivity to RGF1 [44], (Figure 2a). These findings demonstrate the molecular interactions among the RGF peptide, the receptors, and the downstream signaling molecules associated with the receptors.

Furthermore, a recent discovery has uncovered the regulation mechanisms of root meristem development under the RGF1-receptor pathway. The RGF1-RGFR signaling pathway modulates ROS distribution along the developmental zones of the *Arabidopsis* root and enhances the PLT2 protein stability [45]. Genetic evidence also showed that the RGF1-receptor pathway is required to maintain proper gradients of PLT proteins in the proximal meristem [41]. Application of RGF1 peptide to the *rgf1,2,3* ligand triple mutants cause dramatic enlargement of the PLT2 protein localization to a broader pattern than the localization in the triple mutants upon mock treatment; however, the *PLT2* transcripts were comparable between RGF1 and Mock treatment. These results indicate that RGF defines PLT localizations in the root meristem at the protein level instead of the transcriptional level, possibly through the stabilization of PLT proteins [41]. Furthermore, RGF1 treatment does not enhance the broader localization of the PLT2 proteins in the *rgfr 1,2,3* receptor mutant. This indicates that the RGF1-receptor pathway modulates the PLT2 protein stability. Despite the ligand-receptor pairs having been identified, the downstream signaling of RGF1 remained elusive and the molecular mechanisms involved in root meristem development and the PLT1/2 stability were still unknown. Since RGF1 is received in the meristematic zone by the receptors, the meristematic zone-specific transcriptome analysis after RGF1 treatment was conducted to understand the downstream gene in the meristematic zone [45]. The transcriptome analysis detects significant alteration of expression of the genes related to ROS suggesting that RGF1 might signal through a ROS intermediate to control the size of the meristematic zone. Levels of O_2_^•–^ are elevated in the meristematic zone after RGF1 treatment [45]. On the other hand, H_2_O_2_ concentrations are decreased in the elongation and differentiation zone according to RGF1 treatment [45]. Furthermore, pharmaceutical treatments with RGF1 confirm that the alterations of these ROS distributions modulate the meristematic zone size [45]. These experiments demonstrate that RGF1 regulates the meristematic zone size by increasing O_2_^•–^ in the meristematic zone and decreasing H_2_O_2_ levels in the elongation and differentiation zones [45]. These alterations of ROS distributions cannot be detected in the *rgfr1,2,3* triple mutant demonstrating that the RGF1-receptor pathway modulates ROS distributions [45]. The meristematic zone-specific transcriptome analysis revealed that a novel transcription factor, named *RGF1 INDUCIBLE TRANSCRIPTION FACTOR 1* (*RITF1*), is induced in the meristematic zone within 1 h after RGF1 treatment [45]. Previous cell-type and root developmental zone-specific transcriptome analysis showed that the *RITF1* gene is specifically expressed in the meristematic zone compared with the elongation and differentiation zone [46]. The *RITF1* expression is not increased in the *rgfr1,2,3* mutant even after RGF1 treatment showing that *RITF1* is a downstream gene in the RGF1-receptor pathway. Furthermore, overexpression of *RITF1* mimics the phenotypes of the enlarged root meristem size and the alterations of ROS distributions on the RGF1 treatment [45]. The *ritf1* mutants form shorter roots and have a smaller size of root meristem. RGF1 treatment weakly modulates ROS distribution and meristematic zone size in the *ritf1* mutant compared with the wild type. These findings show that *RITF1* is the primary regulator of the alterations of ROS distribution and the root meristem size under the RGF1-receptor pathway. Pharmaceutical analysis showed that the PLT2 protein stability is altered by modulating O_2_^•–^ and H_2_O_2_ levels according to RGF1. These results clearly provide a link between the RGF1 signaling cascade and ROS signals in the regulation of PLT2 protein stability and root meristem size [45], (Figure 2a).

Another example of the tyrosine-sulfated peptides and receptors regulating a part of root development are CIF1 and CIF2 that are expressed in the root stele and perceived by the SCHENGEN3/GASSHO1 (SGN3/GSO1) in the endodermis, where they are required for the formation and maintenance of the contiguous Casparian strip, a barrier allowing for selective nutrients and ion uptake in the root [8,47,48]. Recently, it has been shown that GSO1 and GSO2 require SERK coreceptor for activation [49]. Supporting these findings, the inducible dominant negative *SERK3* line was found that significantly delays Casparian strip formation. In addition, the binding affinity of SERK1 to GSO1 or GSO2 was higher in the presence of CIF3 when compared to CIF1 or CIF2 [49]. These results highlight the critical role of receptor kinase signaling in maintaining endodermis cell identity. The SGN1, a receptor-like cytoplasmic kinase, is required for the formation and integrity of the Casparian strip [50]. The *sgn1* and *sgn3* mutants exhibit a similar defect in Casparian strip formation; however, the cellular localization of SGN1 and SGN3 is different [50]. SGN1, SGN3, RESPIRATORY BURST OXIDASEHOMOLOG (RBOH, an NADPH oxidase) have been reported to play essential roles in a signal module to drive localized ROS production and a precise lignification in plant roots [51]. In vitro kinase assay indicated that SGN1 phosphorylates RBOHD/H and is a direct downstream component of the SGN3/CIF pathway. The pharmaceutical analysis further suggests that NADPH oxidases act downstream in the SGN3/CIF pathway to determine the spatial extent of ROS production [47,51], (Figure 1d).

## 4. The Roles of ROS in Root Development

Growing evidence indicates that redox-dependent regulation may play a critical regulatory role in root development. ROS act as important signal molecules by cooperating with the antioxidants, ascorbate (ASC) and glutathione (GSH). It was shown that ASC oxidase functions in QC cell maintenance via ASC depletion in maize [17]. Another evidence came from research in *ROOT MERISTEMLESS1* (*RML1*, allelic to *CADMIUM SENSITIVE2*), which encodes a *GSH BIOSYNTHESIS* gene that is important for cellular redox homeostasis in meristem maintenance [52,53], (Figure 2a). The *rml1* mutant abolishes cell division in the root thus failing to establish an active root meristem. Exogenous application of GSH was able to recover the root defects of *rml1* and supplementation of GSH inhibitor to the WT causing a phenotype reminiscent of *rml1*, providing a direct link between the *rml1* phenotype and deficiency in GSH biosynthesis. In addition, GSH impairment leads to a cell division block during the G1-to-S transition phase [53]. These results indicate that redox regulation plays an essential role in root meristem maintenance. This is further confirmed by the finding that discrepancies in O_2_^•–^ and H_2_O_2_ distribution in the root tip dramatically affect root growth and differentiation [18]. Both O_2_^•–^ and H_2_O_2_ have distinct roles and accumulation patterns in the root tip; the former is mainly located in the meristematic zone while the latter is situated in the differentiation zone (Figure 2b) and is involved in growth restriction and root hair formation [18].

Another piece of evidence that showed a direct transcriptional link between ROS distribution and root development came from studies of *UPBEAT1* (*UPB1*), a bHLH transcription factor [19]. *UPB1* acts as a pivotal regulator to modulate the transition from cell proliferation to differentiation by directly regulating the expression of a series of peroxidases specifically in the elongation zone. Abolishment of *UPB1* activity interferes with the ROS balance between the meristematic zones and the cell elongation zone thus resulting in a delay in differentiation [19], (Figure 2a). The WT exhibited a shorter root meristem when supplemented with H_2_O_2_ whereas *upb1* mutants did not show a significant change in the meristem size. Application of different peroxidase inhibitors, KCN and salicylhydroxamic acid (SHAM), led to reduced meristem size in *upb1*, suggesting that the regulation of H_2_O_2_ content by *UPB1*-controlled peroxidase is critical for the regulation of the transition from the meristematic zone to the elongation zone [19]. The *upb1* mutant accumulated more O_2_^•–^ in the meristematic and elongation zone than the WT when examined by either nitroblue tetrazolium (NBT) or dihydroethidium (DHE) stain [19]. These results further support the importance of ROS balance for the transition of root development.

The *UPB1* gene and RGF1 peptide control ROS concentrations in different developmental zones. The *upb1* mutant forms a longer root with a longer meristematic zone; however, the mutant does not form extra stem cells. On the other hand, the *rgf* multiple mutants have a shorter root with fewer stem cells. The meristematic zone-specific transcriptome analysis upon RGF1 did not find differential gene expression of the *UPB1* gene [45]. The transcriptome analysis identified the *RITF1* gene as a downstream target under the RGF1-receptor pathway. The *UPB1* gene functions in the elongation zone as a negative regulator of root development; however, RGF1 and *RITF1* in the meristematic zone act as positive regulators. These results suggest that *UPB1* and RGF1-receptors-*RITF1* differently control ROS concentrations in each root developmental zone.

Moreover, the important role of the *GLUTATHIONE REDUCTASE 2* (*GR2*, also named *MIAO*) gene has been demonstrated. GR2 catalyzes the reduction of glutathione disulfide (GSSG) into GSH, to maintain RAM and regulate root growth [54]. The *miao*, a weak mutant allele of *GR2*, exhibits strong RAM defects and root growth inhibition, and supplementation with either GSH or DTT restores the root defects of *miao* mutant in a dose-dependent manner [54]. The expression levels of *PLT1* and *PLT2* were significantly decreased in the *miao* root, corresponding to the shrunken RAM of *miao*. The additive effect of *miao plt1-4 plt2-2*, which displayed severe RAM defects, suggests that the *PLT* pathway and glutathione-dependent redox signaling act in parallel in RAM maintenance [54], (Figure 2a).

In addition, the *Arabidopsis APP1*, which encodes a P-loop NTPase, modulates the local ROS homeostasis to control root stem cell niche identity [55]. The *app1* mutant reduced both H_2_O_2_ and O_2_^•–^ accumulation in the root meristem, and exogenous application of H_2_O_2_ and methyl viologen (MV), which led to overproduction of O_2_^•–^, and recovered the meristematic defects of *app1* [55]. The key transcription factors that define the stem cell niche, *SCARECROW* (*SCR*) and *SHORT ROOT* (*SHR*), are reduced at the transcriptional and post-transcriptional levels in the impairment of *APP1* [55]. These results illustrate that *SCR* and *SHR* are important targets of *APP1*-modulated ROS signaling to control the identity of the root stem cell niche.

Recently, it was reported that *Arabidopsis PROHIBITIN* (*PHB3*) is pivotal in maintaining stem cell niche identity through restriction of the expression of *ETHYLENE RESPONSE FACTOR*s (*ERF109*, *ERF114,* and *ERF115*) that are also a response to ROS in the root [56], (Figure 2a). The *phb3* mutants accumulated higher ROS in line with an increase in NADH dehydrogenase activity, suggesting that *PHB3* is involved in ROS homeostasis [56]. It has been shown that *ERF115* mediates brassinosteroid signaling to control QC cell division through transcriptional activation *PSK5* [35]. It was further shown that *PSK2* and *PSK5* are direct targets of *ERF109* and *ERF114* by Chip-qPCR. These findings imply that *ERF109*, *ERF114*, and *ERF115* mediate *PHB3*-modulated ROS signaling via PSK to regulate stem cell niche maintenance [56].

## 5. The Roles of RALF Peptides and Their Receptors in Root Development

RALF was discovered in tobacco leaf as a cysteine-rich polypeptide that causes an increase in the pH of the medium [57]. The disulfide bridges between Cys-18 and -28 and between Cys-41 and -47 are important for active RALF formation and further cause an inhibitory effect on root growth [57]. The first *Arabidopsis* RALF was isolated based on its ability to induce a rapid elevation in cytosolic calcium [58]. Subsequently more than 30 *RALF* genes were identified in *Arabidopsis* with highly diverse expression patterns and a wide variety of roles [59,60]. Overexpression of *AtRALF1* and *AtRALF23* result in shorter and bushier *Arabidopsis* plants [61,62]. The *AtRALF1* was expressed mostly in the roots and the *atralf1* knockdown plants exhibited increased root length, lateral root number, hypocotyl elongation, and cell length [63]. RALF1 peptide is perceived by FERONIA (FER) (Figure 1a), a member of the *Catharanthus roseus* RLK1-like (*Cr*RLK1L) family that was originally identified as a regulator of the communication between the male and female gametophytes during fertilization [5,64,65]. A phosphoproteomic approach was used to identify phosphopeptides that were significantly perturbed upon RALF1 treatment. FER is one of the candidates that showed an increased abundance of phosphopeptide. The *fer-4* mutant was insensitive to RALF treatment and unable to elicit an increase in cellular Ca^2+^ signaling or inhibit cell elongation. The physical binding of RALF1 to FER was further confirmed by co-immunoprecipitation and in vitro binding assays [5]. It is possible that RALF1 interacts with other receptors since *fer-4* is not completely insensitive to high concentrations of RALF peptide. RALF1 also stimulates the phosphorylation of AHA2 at Ser-899 to inhibit its proton pump activity, causing a reduction in cell elongation [5]. Intriguingly, PSY1 and RALF have opposite effects on plant growth by executing positive and negative regulation, respectively, on the membrane-localized AHA2 [5,38].

The RPM1-induced protein kinase (RIPK) was shown to associate with FER upon RALF1 treatment through MS analysis [66]. The *ripk* mutant displayed the short root hair phenotype and was insensitive to RALF1 treatment similar to *fer-4*; in addition, overexpressed *RIPK* could partially rescue root hair defects in *fer-4* mutant, suggesting that FER and RIPK may work together to control root growth [66], (Figure 1a). The interaction between RIPK and FER was confirmed by a series of biochemical experiments and the interaction was increased upon RALF1 treatment, suggesting that RALF1 ligand binds to FER further recruiting RIPK into the kinase complex and initiating a downstream phosphorylation cascade, resulting in extracellular alkalinization and root growth inhibition [5,66]. Currently, a root growth inhibition assay was carried out by supplementation of 32 synthetic AtRALF peptides. Twenty out of 32 tested RALFs showed a significant root growth inhibition. Surprisingly, *fer-4* mutants were insensitive to 16 out of 20 RALFs that inhibit root length in WT, suggesting that FER is involved in the signaling pathway of most RALFs in root growth inhibition [67].

In addition, FER also interacts with the plant-specific guanine nucleotide exchange factors (RopGEFs), RopGEF4 and RopGEF10, to control ROS-mediated root hair development [68]. Recently, it was shown that the RALF1–FER complex promotes the phosphorylation of EUKARYOTIC TRANSLATION INITATION FACTOR 4E1 (eIF4E1) to regulate the translation of ROOT HAIR DEFECTIVE 6-LIKE 4 (RSL4) and ROP GTPase (ROP2) further determining root hair size and polarity. Moreover, *RALF1* expression is negatively regulated by *RSL4* thus forming a negative feedback loop to fine-tune root hair development [69,70], (Figure 1a). RALF34 peptide is perceived by another *Cr*RLK1L member, THESEUS1 (THE1), to fine-tune pericycle cell division patterns during lateral root initiation [71,72]. The *ralf34* and *the1* mutants displayed comparable lateral root phenotypes and *the1* mutant is insensitive to RALF34-induced root growth inhibition and surface alkalinization [71], (Figure 1c). FER also plays a key role in maintaining the integrity of the actin cytoskeleton to mediate the PIN-FORMED 2 (PIN2) protein polar localization that contributes to the lateral root development and gravitropic responses [73].

## 6. Conclusions and Perspectives

The peptide hormones play versatile roles in coordinating diverse aspects of plant root development, including root meristem maintenance, cell expansion, lateral roots, and root hair formation. Despite a growing number of studies on the roles of signal peptides in plant root development, the specific functions of many of these peptides are largely unknown. Given the numerous secreted signal peptides, our knowledge regarding their receptor recognition mechanisms, intracellular signaling pathways and downstream ligand-receptor pairs is still limited. Therefore, continued efforts should be made to unravel the molecular functions of the peptide hormones.

The major bottleneck in characterizing the potential roles of CLE peptides is genetic redundancy since most of the *cle* mutants did not display distinct phenotypes. Several questions need to be addressed, such as the specific functions of each CLE peptide; the receptors they interact with under certain environmental stimuli or developmental stages; along with how CLE peptides can be modified and whether this affects their activity. PTM is a very important process for the maturation and activation of a small peptide. The hydroxyproline of the CLV3 peptide is arabinosylated during processing [74]. The modification of CLV3 strongly enhances the inhibition activity of SAM and the binding to the receptor. However, the modification of only a few of the 32 CLE peptides has been identified. Previously, chemically synthesized CLE peptides without modification underwent root inhibition assay. It is possible that the chemically synthesized CLE peptide may bind to multiple receptors. The receptor mutants are resistant to the inhibition activity of many chemically synthesized peptides; however, they may show resistance to the specific mature modified form of CLE peptide. Further understanding of the modification of the CLE peptide and the interaction between the CLE peptide and receptors will uncover the roles of CLE peptides in root meristem development.

The RGF peptides positively regulate root meristem development by enhancing stability of PLT1/2 protein via ROS. However, it is still unclear what kinds of regulatory components are involved in the RGF-RGFR-RITF1 signaling cascade for regulating ROS and how ROS regulates PLT1/2 protein at the translational level. We will get a comprehensive understanding of how root cells communicate with proximal cells via signaling peptides. ROS have emerged as essential regulators in maintaining cell division and differentiation thus controlling root growth. However, further study is needed to clarify what exactly ROS do and how they regulate root master regulators PLT1/2. It is possible that ROS modulate PLT1/2 through PTMs, such as phosphorylation, ubiquitinylation, or sulfenylation. Particular attention should be paid to identifying the receptor(s) that transduces the ROS signal. Future research should also aim to further elucidate the interplay between peptide hormones and ROS in root development.

The RALF peptides affect cell expansion and plant development via Ca^2+^ signaling, MAP kinase signaling, and pH modulation; however, the order of specific events and interactions through which RALFs regulate pH are still unknown. Also, it should be considered that the pH may affect the binding affinity between some RALFs and their receptors. To date, only a few RALFs have been demonstrated to interact with their *Cr*RLK1Ls. It is essential to decipher yet unexplored RALFs and their receptors further to address the mechanism underlying the specificity of RALF-receptor associations and how individual RALFs coordinate numerous signaling complex and specific responses.

## Figures and Tables

**Figure 1 genes-12-00022-f001:**
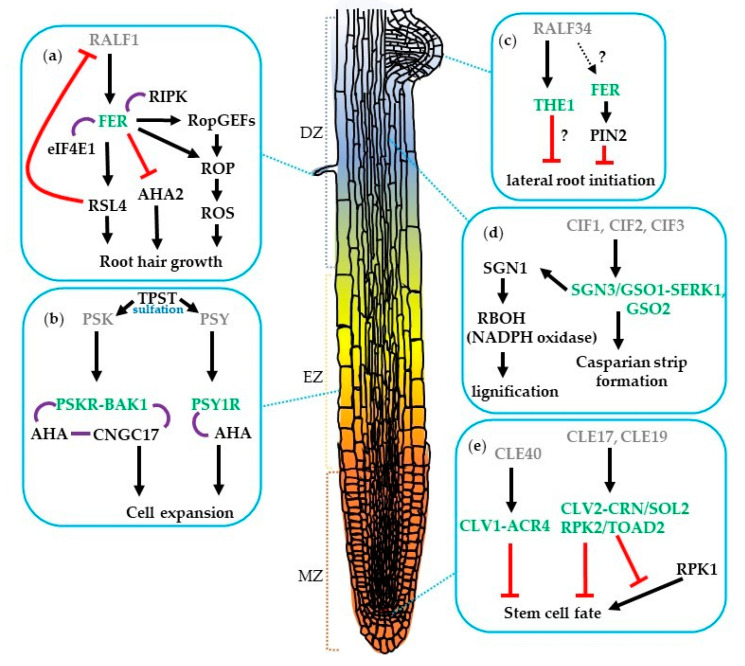
Regulation of root development by (**a**,**c**) RALF, (**b**) PSK and PSY, (**d**) CIF, and (**e**) CLE peptide signaling pathways. The small peptides (grey) and their receptors (green) involved in lateral root initiation, Casparian strip formation, root hair growth, root cell expansion, and root stem cell fate in *Arabidopsis*. Arrows indicate positive regulation; red bars show negative regulation; purple lines represent association; and question marks indicate unknown mechanism. MZ, meristematic zone; EZ, elongation zone; DZ, differentiation zone.

**Figure 2 genes-12-00022-f002:**
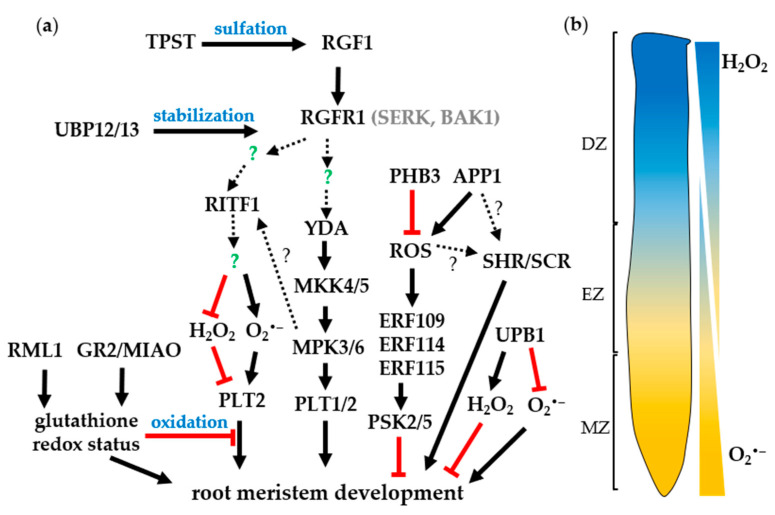
The signaling cascade of (**a**) RGF1-RGI1 and ROS signals to regulate root meristem development. SERK and BAK1 act as co-receptors for RGI1. Arrows indicate positive regulation; red bars mean negative regulation; green question marks indicate unknown players; and broken lines with question marks represent further experiments are needed. (**b**) The H_2_O_2_ and O_2_^•–^ distribution in root. MZ, meristematic zone; EZ, elongation zone; DZ, differentiation zone.

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
