# Peer review of "The Roles of Peptide Hormones and Their Receptors during Plant Root Development"

_genes, 2020, doi:10.3390/genes12010022_

Round 1

Reviewer 1 Report

In their interesting review article the authors address an important, yet somewhat underestimated issue in plant develoment. The role of peptide hormones and their close relationship with reactive oxygen species in establishing signal cascade modules is explored in this paper. Substantial novel data published since the Matsubayashi 2006 review article (Ann Rev Plant Biol) are summarized here. It was about time. The manuscript is clearly written, in spite of the complexity of the topic, and also provides useful hints for further research. I have only a few minor suggestions for further improving this useful contribution.

  1. Line 51: The authors start the paragraph stating that ROS are undesirable byproducts. This is the "old view", and it is misleading. As the paragraph goes on, the signaling role of ROS is introduced, but the feeling that ROS are something you don't want to have around is still there. It should be considered that not only ROS spill from several metabolic processes, but they are also purportedly produced by NADPH oxidases. The excellent 2017 TIPS paper by Ron Mittler, with its revealing title ("ROS are good") could be mentioned  to introduce the section. Later on in the manuscript (line 387) it is clearly stated that ROS have emerged as essential regulators of development. Essential regulators are not supposed to be byproducts.
  2. Ascorbate is not mentioned in the list of potential players. The Feldman group published some papers discussing the possible involvement of ascorbate oxidase in maintaining the stem cell niche in the RAM. Glutathione, discussed in the manuscript in connection with RML1 and GR2, strictly relates to ascorbate and dehydroascorbate. An additional intriguing clue is the hydroxylation of proline residues in the CLE peptides (a process requiring ascorbate). Maybe a quick mention could be useful.
  3. As a minor remark, it is advisable to check very carefully for the use of italics to indicate genes. Line 63: CLE genes should go italics. Please double check throughout the manuscript.        

Author Response

Dear reviewer,

Reviewer 2 Report

A nice short review on the role of peptide hormones, their receptors and ROS in plant root development.

General comments

A good introduction to the topic.

In each section you mention a lot of genes/peptides/receptors etc, which can make the reader get lost quite easily. To try and guide the reader through this, I would suggest making sure that every gene/peptide/receptor mentioned in the text is also included in one of the figures. I will include examples below of where I think this could be done.

You often refer to the figures at the end of a paragraph. Can I suggest referring to the figure at the first mention of something relevant to the figure as this will help the reader understand the text in the paragraph.

A few minor grammatical errors throughout.

Specific comments

Lines 40-41– make sure you italicise any gene/transcript names e.g. here, WUSCHEL and CLV3 should be italicised as you are referring to gene expression. This applies throughout the whole document.

Lines 46-47 – Can you refer to Figure 1 here and indicate in the figure legend where the three zones are located?

Line 50 – none of these references refer to the role of small peptides in root growth rate, please add an appropriate reference

Lines 55-59 – the wording could be improved here to make it clear that singlet oxygen and superoxide are the small molecules you are referring to.

Lines 59-61 – need to briefly mention that you are going to discuss ROS in the manuscript

Line 67 – specify what type of cle mutants e.g. knockout/knockdown?

Line 68 – you mention that there are two functional types of CLEs, but only name one. Can you briefly mention the other type too?

Lines 70-71 – Include a reference for this sentence and/or make it clear that the one example is CLE40 and CLV1/ACR4

Line 73 – change ‘sensitive against’ to ‘sensitive to’

Line 76 – Do you mean mutants of putative A-type CLE peptide receptor genes? Make sure it is clear what receptors you are referring to.

Lines 77-78 – Wording is not clear here. Do you mean that sol2/crn can suppress the short-root phenotype observed in CLE19 overexpressing lines?

Line 79 – If my interpretation of the text and Figure 1 is correct, I would suggest rewording this sentence to say ‘…SOL2/CRN and CLV2 work together both dependently and independently of CLV1 in the SAM’

Lines 80-82 – refer to Figure 1 here.

Lines 82-83 – cle19 is the only mutant mentioned in the paragraph, so I would suggest changing the text to ‘However, cle19 mutants and the sol2/crn and clv2 receptor mutants…’, unless you are referring to more than one cle mutant, in which case state what they are.

Lines 90-91 – Add RPK1 to Figure 1

Figure 1 – I suggest labelling each box in the figure A-E and then referring to each specifically in the text e.g. line 75 refer to Figure 1E.

Lines 104-105 – these first two sentences belong somewhere in the previous section. Then change wording of lines 105-106 to something like ‘While CLE peptides negatively regulate meristem development, tyrosine-sulfated peptides positively control root meristem development’

Lines 106-108 – change wording to ‘PSKs, PSHs, and RGFs rely on tyrosine sulfation, a post-translational modification catalyzed by the product…’ Also include a reference here. I would also add TPST to figure 1 and then refer to the figure in this sentence.

Line 112- remove the word further as this is the first piece of evidence you have provided for this.

Lines 114-119 – refer to Figure 2 somewhere here as I spent awhile searching for RGF1 in Figure 1, and only later realised that you had included it in a separate figure.

Line 125 – There needs to be better linkage when you mention PSY1 e.g. ‘PSY1, on the other hand, is perceived by…’. This applies in other sections of the manuscript as well, as you sometimes jump from one idea to the next without any linkage which can make it difficult for the reader to follow.

Lines 142-145 – these two sentences could be combined into one as they are essentially saying RGF1-5,8 do the same thing.

Lines 199-200 – move to the end of the previous paragraph and add a reference to Figure 2.

Lines 232-234 – you say that there is a clear link between RGF1 and PLT2, but in Figure 2 there is a question mark over this link.

Line 241 – Hence is not the correct word to use here. What is said in this sentence is not a conclusion from the previous sentence.

Line 243 – this is the first time you have used the RK abbreviation, so please spell it out in full

Lines 248-249 – in vitro should be italicised.

Lines 250-252 – NADPH oxidases need to be added to Figure 1.

Line 254 – add an introductory sentence or two on ROS and root development.

Lines 254-266 – can you add this information into one of the figures or make a new figure?

Line 258 – suppletion is the wrong word here – I think you might mean supplementation?

Line 260 – again hence is the wrong word to use here.

Lines 299-306 – can this information also be added to one of the figures?

Lines 307-315 – ROS and PSK2/5 are missing from the PHB3 pathway in Figure 2.

Line 324 – Atrafl1 should be in lower case italics if you are referring to the mutant.

Line 328 – move the Figure 1 reference to directly after ‘RALF1 peptide is perceived by FERONIA (FER) (Figure 1), …’.

Line 334 – add AHA2 to Figure 1 downstream of RALF1?

Lines 347-353 – should there be a direct arrow from FER to ROP? Can you reword this section to make it clearer? I felt like the text and Figure 1 information did not entirely match.

Lines 357-359 – Is this downstream of RALF34? Can you add PIN2 to Figure 1 as well?

Conclusion and Perspectives – good first paragraph. Is there any reason why you only discuss CLE and RGF peptides in this section and not the other peptides mentioned throughout? I feel like these last two paragraphs might belong better within Section 2 and 3 and then have a more general conclusion section that summarises all the peptides as well as ROS?

Author Response

Dear Reviewer,

Sincerely,

Masashi Yamada
